# Influence of Reactive Amine-Based Catalysts on Cryogenic Properties of Rigid Polyurethane Foams for Space and On-Ground Applications

**DOI:** 10.3390/ma16072798

**Published:** 2023-03-31

**Authors:** Vladimir Yakushin, Maris Rundans, Malgorzata Holynska, Beatrise Sture, Ugis Cabulis

**Affiliations:** 1Latvian State Institute of Wood Chemistry, Polymer Laboratory, Dzerbenes Street 27, LV-1006 Riga, Latvia; 2ESA/ESTEC, Keplerlaan 1, 2201 AZ Noordwijk, The Netherlands

**Keywords:** rigid polyurethane foams, cryogenic insulation, environmentally friendly catalysts and blowing agents

## Abstract

Rigid polyurethane (PUR) foams have outstanding properties, and some of them are successfully used even today as cryogenic insulation. The fourth-generation blowing agent Solstice^®^ LBA and commercial polyols were used for the production of a low-density cryogenic PUR foam composition. A lab-scale pouring method for PUR foam preparation and up-scaling of the processes using an industrial spraying machine are described in this article. For the determination of the foam properties at cryogenic temperature, original methods, devices, and appliances were used. The properties at room and cryogenic temperatures of the developed PUR foams using a low-toxicity, bismuth-based, and low-emission amine catalyst were compared with a reference foam with a conventional tin-based additive amine catalyst. It was found that the values of important cryogenic characteristics such as adhesion strength after cryoshock and the safety coefficient of the PUR foams formed with new reactive-type amine-based catalysts and with the blowing agent Solstice^®^ LBA were higher than those of the foam with conventional catalysts.

## 1. Introduction

So far, different methods of hydrogen storage for ground and aerospace vehicles have been proposed and investigated in different studies [1,2,3,4,5,6,7,8,9], and it has been shown that the most favorable type of tank system depends highly on the kind of vehicle and operation [10]. When using hydrogen as an aviation fuel, whether at subsonic or hypersonic speeds, lightweight, insulated cryogenic tanks are a key technical enabler as the low density of H_2_, even when stored as a liquid, leads to a tank volume that is about four times higher than that of an equivalent hydrocarbon fuel in spite of its reduced mass [11,12]. The high temperatures encountered at hypersonic speeds make tank construction potentially even more challenging as the insulation system has to be designed to withstand a higher thermal load for a comparatively long time.

Rigid polyurethane (PUR) foams have outstanding properties, and some of them are successfully used even today as cryogenic insulation. Polyurethane foams are also regarded as the main material to be used in the future for liquefied natural gas (LNG) tankers, also as cryogenic insulation in different projects of ground, air, and space vehicles. There are a whole range of examples of the use of plastic foams as cryogenic insulation [13,14], although manufacturers and developers of insulation do not always provide information on the chemical nature and properties of the used foam material.

Polyurethane foams possess many advantageous properties, such as low thermal conductivity, a light weight, low water absorption/permeability, and dimensional stability. In particular, polyurethane foams do not change in volume along temperature gradients; in other words, they exhibit significant size stability [15,16]. Polyurethane foams are usually created by foam expansion as a result of the reaction between polyols (component A) and isocyanate (component B). For practical reasons, component A usually contains not only polyols but also a combination of flame retardants, surfactants, catalysts, and a blowing agent.

A traditional catalyst package for the development of PUR foams contains several chemicals intended to improve the blowing or gelling or a balance of both processes during foam production. The majority of these are ‘non-reactive’ catalysts, which are recognized as a health hazard due to their potential volatility, thus evaporating and interacting with the surrounding environment. To negate this, several new ‘reactive amine catalysts’ that, unlike their predecessors, react with isocyanate or polyol molecules and incorporate into the polymer matrix have been developed. TOSOH company [17] tested several novel triethylenediamine (TEDA)-based non-volatile reactive amine catalysts to be used in conjunction with metalorganic catalysts for the production of elastic PUR foams, finding that RZETA, a hydroxymethyl-TEDA variant showed better gelling activity in comparison to other reactive amine catalysts. Sikorski et al. [18] replaced traditionally used non-reactive TEDA with reactive catalysts PC CAT^®^ HPI and API, reducing volatility caused by silicone surfactant reactions. Casati et al. [19] investigated several reactive amine catalysts in combination with so-called “active polyols” from The Dow Chemical Company, concluding that the combination of both or the use of active polyols alone helps in the reduction of volatile components during the preparation of elastic PUR foams. Muuronen et al. [20] used a novel computational method to successfully predict the catalytic activity of seven different N,N-dimethyl group-containing catalysts; they found that aliphatic heterocycle-containing catalysts are comparable to tertiary amine catalysts, with catalytic activity increasing with decreasing heterocycle ring size, suggesting pyrrolidine derivatives as high-catalytic-activity alternatives without the risk of formaldehyde formation. Zimmerman et al. [21] used a combination of only hydroxyl-containing reactive amines (JEFFCAT^®^ ZF-10 and ZR-70) in order to successfully prepare elastic PUR foams for car seats. The high catalytic activity and low emission characteristics of the same catalyst group were later proven by Chaffanjon et al. [22] using real-time FTIR kinetic analysis. An alternative approach in novel reactive catalyst preparation has recently been shown by Stridaeng et al. [23] who prepared copper and zinc acetate–ethanolamine complexes and used them for the preparation of rigid PUR foams; both catalysts showed complete reaction activity, with longer gel and rise times in comparison to N,N-dimethylcyclohexylamine.

The traditionally used metalorganic tin-based catalyst for urethane reactions also raises concerns due to its potentially negative environmental impact. Pretti et al. [24] conducted ecotoxicological studies of bismuth-based and tin-based catalysts and found that the tin-based catalyst was toxic to several tested marine specimens, while in contrast the bismuth-based catalyst was non-toxic to any of the tested species. As a result, the use of a bismuth-based catalyst over a tin-based one has at least a clear ecological advantage. Khezraji et al. [25] replaced a traditionally used tin-based catalyst with a non-toxic bismuth-based catalyst for elastic PUR foam development and found that the bismuth-based catalyst showed a higher isocyanate conversion rate than the tin-based catalyst, leading to improved mechanical properties.

Up until the early 2000s, the main chemicals used as refrigerants, propellants, solvents, and of course blowing agents for polymeric foam production were hydrofluorocarbons (HFCs). It was discovered that, despite being environmentally friendly to the ozone layer, their influence on global warming potential (GWP) spelled disaster. As a result, Kyoto Protocol, pursuant to the United Nations Framework Convention on Climate Change (UNFCCC), set binding targets for greenhouse gas emissions based on calculated equivalents of carbon dioxide, methane, nitrous oxide, HFCs, PFCs, and sulfur hexafluoride [26]. The limit for the use of such substances is their GWP value, composed of the combination of the radiative forcing and atmospheric lifetime over the evaluated time frame (i.e., over 100 years) relative to what carbon dioxide creates [27]. The European Parliament has adopted “F-Gas regulation” for the implementation of the Kyoto Protocol which deals with fluorochemicals with a GWP value over 150 for a 100-year projection. Gradual phase-out and emission reduction down to 99% according to a schedule is ongoing. The current schedule [28] foresees near-complete phase-out of all HFCs as of 2023. As a result, hydrofluoroolefins (HFOs) have been proposed for HFCs’ replacement. With an unsaturated carbon–carbon bond within their structure, HFOs’ atmospheric lifetime drastically decreases (due to rapid decomposition), yielding GWP values way below the 150 mark; the presence of fluorine atoms also provides fire resistance and ODP values of, or close to, zero. Thus, current “fourth generation” blowing agents are mainly composed of various HFOs, as well as certain ether, aldehyde, and alkane alternatives. The presence of an unsaturated bond in HFOs’ structure creates certain chemical challenges as it means previously used catalysts and surfactants may react with the new blowing agent, yielding PUR foams with deteriorated physical–mechanical properties. To combat this, a line of new amine catalysts for PUR systems have been developed [29]. 

Modern PUR foam formulations for cryogenic spray application must be developed in such a way as to have an outstanding combination of mechanical, thermal, and physical properties. Due to the ease of physical and mechanical deviations that may occur during either spray application or afterwards during curing, the chemical interaction of raw components (polyols, catalysts, surfactants, blowing agent) must be well studied; therefore, PUR foam systems have a complex development with many complicating factors requiring extensive testing and characterization [30]. The current trends, which are in line with European and global environmental policy, are the use of renewable raw materials in PUR research and development [31] and industry [32], and also in the development of cryogenic insulation [33,34]. Non-isocyanate polyurethane foams are also being developed for use in cryogenic insulation [35]. In previous years, several researchers have experimentally tested the mechanical properties of PUR foams under various environmental conditions and material states, for example, temperature, pressure, strain rates, etc. The mechanical features of PUR foams were investigated at various material densities [36] and quasi-static and high strain rates [37,38]. Moreover, the microscopic characteristics [39], cushioning properties [40], energy absorption [41], thermal conductivity [42], and static compressive loading [43] of PUR foams were investigated for identifying their material characteristics. However, the operating environment of a launch vehicle is a combination of rapid temperature and atmospheric changes; aerodynamic, acoustic, and mechanical loads; and other conditions. The cryogenic tank’s foam insulation is therefore subjected to a number of potential structural problems, including potential debonding from the substrate, interlaminar adhesion failure, mechanical (tensile, compressive) failure, etc.

This study presents the results of a comparative study of several of these parameters at normal and cryogenic conditions for rigid PUR foam developed using fourth-generation blowing agents and appropriate reactive amine-based catalysts.

## 2. Materials and Methods

### 2.1. Materials

For PUR foam production, polyether and polyester polyols Lupranol 3300, Lupranol 3508/1, and Lupraphen 1901/1 were purchased from Ludwigshafen, Germany. In addition, diethylene glycol, technical grade, from Sigma-Aldrich (Steinheim, Germany); reactive IXOL B 251 from Solvay Fluor (Hanover, Germany) and additive TCPP (tris-(1-chloro-2-propyl) phosphate) (Albemarle, Louvain-la-Neuve, Belgium) as flame retardant; and Silicone L-6915LV (Momentive Performance Materials, Leverkusen, Germany) as surfactant were used for polyol mixture preparation. Catalyst tin butyl dilaurate (trade name Kosmos 19) from Sigma-Aldrich (Steinheim, Germany), Dabco MB20, Polycat 5, Polycat 203, and Polycat 218 (Evonik GmbH, Essen, Germany) in various combinations were used as catalysts. Polymeric 4,4′-methylene diphenyl isocyanate (pMDI) Desmodur^®^ 44V20L (Covestro AG, Lervekusen, Germany) with an NCO group content of 31.5% and an average functionality of 2.7 was used as an isocyanate component in polyurethane formulations. Average molar mass of the chains between cross-links (Mc) was used as a basic characteristic for composition calculations and optimization [44,45]. In this study, cryogenic insulation compositions had an M_c_ of 550 g/mol. A combination of two blowing agents, a chemical blowing agent of water (water in polyols and added water), and a physical blowing agent HFO-1233zd-E under the trade name Solstice^®^ LBA (Honeywell Fluorine Products Europe B.V., Weert, The Netherlands) was used.

### 2.2. Methods

#### 2.2.1. Preparation of PUR Foam Samples

The research was carried out in 2 stages: (1) system optimization with the pouring method; (2) methods of up-scaling using an industrial spraying machine. The recipes of the PUR compositions used in the study are given in Table 1 and Table 2.

At first, size samples (cup tests) were obtained using the universal foam qualification system FOAMAT 285, which measures different foaming parameters of start time, gel time, and tack-free time.

To determine the effect of the new blowing agent content on the density and adhesive strength, PUR foam blocks were made by pouring. For this, a polyol mixture with a reduced concentration of the amine catalyst (0.5 part by weight (pbw)) was used. Pouring free-rise PUR foam blocks were prepared using a laboratory mixer with a stirrer speed of 2000 rpm and open molds with a size of 250 × 250 × 100 mm. Poured foam was applied on aluminum plates of 40 × 40 × 4 mm, preliminarily abraded with sandpaper. The temperature of the aluminum plates was 22 °C.

For the spraying of PUR foam panels, the same polyol mixture with an increased content of the amine catalysts was used. The blowing agent was added in such an amount (0.35 mol) as to obtain panels with an apparent density of about 35 kg/m^3^. For the spraying of foam panels, a high-pressure GlasCraft MH VR dispensing system and a spray gun (Probler P2 Elite) were used. The polyol and the isocyanate components’ temperatures were 40 °C, and the working pressure of the components was 120–140 bar. Polyurethane foam panels were spray-applied on aluminum sheets covered with a release agent. For the adhesive test, PUR foam was spray-applied on the same aluminum plates of 40 × 40 × 4 mm. The temperature of the aluminum sheets and plates was 22 °C. The thickness of spray-applied panels and poured blocks was 50–60 mm. 

#### 2.2.2. PUR Foam Testing

PUR foam samples for testing were cut out from the core of the poured blocks and sprayed panels. For mechanical tests of PUR samples, the Static Materials Testing Machine Zwick/Roell Z010 TN (10 kN) (Zwick GmbH & Co, Ulm, Germany) with the Basic program testXpert II was used. For the test at cryogenic temperature (77 K), the testing machine was equipped with the original cryostat and block for the temperature regulation (Figure 1a). 

For the tensile test at cryogenic temperature, a special appliance and foam rings with an inner diameter of 43 mm, outer diameter of 53 mm, and width of 13–14 mm were used (Figure 1b). Rings were cut out in-plane perpendicular to the foam rise direction. The reliability of this method of foam tensile testing, analogous to ASTM D 2290, is described in [46,47,48]. 

The adhesive strength of PUR foam to aluminum was measured as tensile bond strength between polyurethane foam and aluminum according to EN 1607. Aluminum plates (40 × 40 mm) with applied foam were cut off from poured blocks or sprayed panels on aluminum plates. The thickness of foam after cutting off for all samples was 20 mm. Tensile bond strength was determined at room temperature and after the cryoshock test (immersion of aluminum samples with applied insulation in liquid nitrogen and subsequent exposure of it for an hour). After this test, samples were glued to aluminum plates (40 × 40 mm) with adapters for fixing in the test machine (Figure 1c). Each mechanical testing series was made using 8 samples.

The value of the contraction of foam at cooling from room to cryogenic temperature was determined using the vertical thermomechanical analyzer TMA PT1600 (Linseis GmbH, Selb, Germany). For this test, specimens of 4 × 4 × 20 mm were used. Ring-like specimens were cut off in-plane perpendicular to the foam rise direction. PUR specimens were cooled at a rate of 3 K/min. The analyzer made it possible to determine the contraction only in the temperature range from 295 K to 105 K. For calculation of the safety coefficient, the required contraction in the temperature range from 295 K to 77 K was determined by extrapolation of experimental dilatometric curves.

The coefficient of thermal conductivity was carried out according to ISO 8301:1991 using Linseis Heat flow meter 200 (Linseis, Selb, Germany), The sample (dimensions, 200 mm × 200 mm × 50 mm) was inserted between two plates (top plate temperature, 20 °C; bottom plate temperature, 0 °C), and the coefficient of thermal conductivity was measured at +10 °C. 

Fourier transmission infrared spectroscopy (FTIR) analysis was performed using a Thermo Scientific Nicolet iS50 spectrometer (Waltham, MA, USA). FTIR spectra were obtained using absorbance at a resolution of 4 cm^−1^ with 32 scans.

## 3. Results and Discussion

Among the various criteria for the stability of PUR foam insulation at cryogenic temperatures, two should be distinguished. The first of these is the high adhesive strength of the foam to the substrate material, which is maintained even at cryogenic temperatures. In this study, it was evaluated as the tensile bond strength of foam applied on an aluminum plate. 

The second criterion is the ability to withstand the formation of cracks in the insulation when it cooled to cryogenic temperatures, due to the difference in the coefficients of thermal expansion of the substrate material and insulation. The ability of materials to resist the formation of cracks at cryogenic temperatures can be estimated from the maximum thermal stresses that occur in the foam on the surface of the insulated substrate. The safety factor in this case [49] is measured as the ratio of the strength of the foam at cryogenic temperature to the resulting thermal stresses:(1)$${K_{\sigma }=\sigma }_{t}\cdot (1-\nu )/(E\cdot \Delta T)$$
where: *σ_t_*—tensile strength at cryogenic temperature, MPa;*E*—tensile modulus at cryogenic temperature, MPa;Δ*T*—temperature difference, K;*ν*—Poisson’s ratio at cryogenic temperature.

Another way to assess the reliability of cryogenic insulation is the criterion of maximum deformations. In this case, this ability can be estimated as the ratio of the elongation at break of the polyurethane foam at cryogenic temperature to the contraction of the foam when it is cooled from room temperature to cryogenic temperature (safety coefficient $K_{\varepsilon }$) [50]:(2)$$K_{\varepsilon }=\varepsilon_{77}/\Delta l_{295-77}/l_{295}$$
where: $\varepsilon_{77}$—elongation at break at 77 K, %;$\Delta l_{295-77}$—contraction of foam at cooling from 295 to 77 K, %;*l*_295_—length of sample at room temperature, mm.

It was according to this criterion that the studied materials were evaluated. This simplified formula does not take into account the difference in the thermal expansion coefficients of the foam and the substrate material. Given this difference, the value of the coefficient will be slightly higher.

### 3.1. Preliminary Cryogenic Tests with Pouring Compositions

The properties of PUR foams are determined by both the properties of the polymer matrix and the density. To determine the dependence of the adhesive strength of the foam on the density, foam of different densities was applied to aluminum plates by pouring. Both Solstice^®^ LBA and its combination with water were used as a blowing agent to vary the density of the polyurethane foam CRYO_p. The total content of physical (Solstice^®^ LBA) and chemical (water) blowing agents was estimated in moles (Table 3). The properties of CRYO_p with Solstice^®^ LBA obtained with the additional use of 0.5, 1.0, 1.5, and 2.0 pbw of water are indicated in the figures as 0.5 W, 1.0 W, and so on. The foam prepared without the use of additional water during foaming is indicated as 0.1 W in the figure, because the total moisture content of the polyols was 0.1 pbw. The tin-based catalyst Kosmos 19 and the amine catalyst Polycat 5 were used as catalysts in the pouring CRYO_p formulations.

The apparent density of the obtained foam, depending on the total amount of blowing agents, is described quite accurately by one approximation curve, regardless of the amount of water in the polyol composition (Figure 2a). Naturally, with an increase in the total content of the foaming agent, the apparent density of the foam decreases.

The total amount of blowing agent has the same effect on adhesive strength. However, the values of the adhesive strength of the foams obtained with the use of a minimum amount of additional water (0.5 W) and without it (0.1 W) are slightly higher than the approximation curve on the figure. The adhesive strength values of the foams obtained with the use of more additional water lie slightly lower than the approximation curve (Figure 2b). The same is observed in the figure of the dependence of adhesive strength on the density of the foam (Figure 2c).

Such a small difference in the adhesive strength of the composition with a higher content of water can be explained by the presence in the resulting polyurethane of, in addition to urethane groups, urea, biuret, and allophanate groups. These compounds are formed as a result of a secondary isocyanate reaction with amine, urea, and urethane at the polyurethane foaming with the water [51].

The dependence of the adhesive strength of the foam after cryoshock on the density is shown in Figure 2d. The values of adhesive strength after cryoshock were slightly less than the initial adhesive strength of the foam. However, no delamination of the foam after the cryoshock test was observed in all cases. Like the initial adhesive strength, the strength of the foam after cryoshock decreased with decreasing density.

### 3.2. PUR Foam Scale-Up Production Using Industrial Spraying Machine and Cryogenic Tests

The foams were sprayed using a variable-ratio high-pressure GlasCraft MH VR dispensing system that allowed polyurethane compositions to be sprayed with varying Solstice^®^ LBA content at a given NCO index. This high-pressure machine also made it possible to evaluate the effect of small changes in the isocyanate index (within 1.1–1.2) on the properties of the foam. Isocyanate–polyol and isocyanate–water reactions are both catalyzed by amines. Using tertiary amines has some disadvantages, including their pungent fishy smell and high volatility. Reactive catalysts have emerged as a result of growing environmental concerns about reducing emissions of volatile organic compounds (VOCs). Recent Negligible-Emission-grade catalysts from the company Evonik have lower emissions compared to traditional amines, lowering employee and consumer exposure to VOCs. We applied catalysts from the Negligible Emission grades in our research.

The following catalysts were used in spray compositions:Kosmos 19, a tin-based strong gel catalyst that was used as a reference metal-based catalyst;Dabco MB 20, a bismuth-based carboxylate that is an alternative to tin-based catalysts in rigid foam systems;Polycat 203, recently developed and patented, is a low-water-containing, amine-based catalyst with outstanding stability in formulations that contain HFO blowing agents. In comparison to conventional catalysts, Polycat 203 is a reactive amine catalyst and can facilitate a reduction in amine emissions during spraying;Polycat 218, recently developed and patented, is a relatively HFO-stable blowing reactive amine catalyst, designed to complement Polycat 203 in formulations containing HFO;Polycat 5, a conventional additive blowing catalyst that was used as a reference amine catalyst.

The preliminary experiments have shown that in order to ensure good adhesive strength of the polyurethane foam to aluminum during spraying at room temperature (without heating the metal), it is necessary to use at least 6 pbw of the amine catalyst. The influence of the catalyst on the foaming parameters is shown in Table 4, where it can be seen that at concentrations of 6 pbw, the foaming parameters reach the optimum for spray recipes: start time of 3–5 s; rise time of <30 s.

Initially, PUR formulations using HFO as a blowing agent were developed and tested at room temperature [29], with approximately the same apparent density for pouring and spray foams, 34–36 kg/m^3^. Both foams had a closed-cell structure. The volume content of the closed cells in pouring and spray foams was practically the same, namely, 95 vol%. It was found that at practically the same density, the initial value of the thermal conductivity coefficient λ_10_ of the pouring foam was higher by 10% than that of the spray foam (17.1 mW/m·K versus 15.4 mW/m·K). Due to the much smaller cell sizes, the spray PUR foam had a much lower thermal conductivity. This is a direct consequence of the decrease in the radiative component in the overall thermal conductivity of the foam [52].

Better results for foam with an apparent density of 35 kg/m^3^ were found for compositions with Solstice^®^ LBA without the use of an additional blowing agent of water. The properties of the foams obtained with the most successful combinations of the mentioned catalysts are shown in Table 5 and Figure 3, Figure 4 and Figure 5. These combinations of gel and blowing catalysts in the sprayed PUR compositions are listed in Table 2.

The adhesive strength of all sprayed foams was tested after the cryoshock test. It was found that with an increase in the isocyanate index, the adhesive strength of the PUR foam after cryoshock increases and for most compositions reaches its maximum at an index of 1.2. A possible reason for this increase in adhesive strength may be the more complete curing of PUR and a slight increase in the cross-link density of it with a slight increase in the isocyanate index. A similar increase in adhesive strength proportional to the cross-link density was noted in [53].

The values of the properties of CRYO_spr_3 were lower than those of the similar CRYO_spr_4 due to a lower content of gel catalyst Dabco MB-20. CRYO_spr_4, containing 0.2 pbw of the catalyst MB-20 and 6 pbw of the catalyst Polycat 218 at the NCO/OH ratio 1.2, had the highest adhesive strength value (Figure 3a). It was higher than the adhesive strength of CRYO_spr_1 with the conventional reference catalysts Kosmos 19 and Polycat 5.

The typical appearance of the specimens after cryoshock and the tensile bond strength test is shown in Figure 3b. As a rule, some of the specimens had an adhesive failure mode, and some had a cohesive one. But in all cases, the foam film remained on the metal surface.

The properties of the foam at cryogenic temperature are shown in Figure 4, Figure 5 and Figure 6. The tensile strength of PUR foam mostly, with rare exceptions, increased with an increase in the isocyanate index for all compositions (Figure 4a). The highest tensile strength values were found for all PUR compositions at the maximal NCO/OH ratio. This effect, as in the case of adhesion, can certainly be facilitated by more complete curing of PUR and a slight increase in the cross-link density of it with a slight increase in the isocyanate index. CRYO_spr_1; CRYO_spr_2; and CRYO_spr_4 had approximately equal maximal values of tensile strength at cryogenic temperature.

PUR foam elongation at break at 77 K, like tensile strength, mostly, with rare exceptions, increased with increasing isocyanate index for all compositions (Figure 4b). The highest elongation values were found for CRYO_spr_1 with the conventional catalysts Kosmos 19 and Polycat 5. However, the elongation of CRYO_spr_4 with 0.2 pbw of the catalyst MB-20 and 6 pbw of the catalyst Polycat 218 is only slightly inferior. 

The contraction of PUR foams upon cooling from room temperature to liquid nitrogen temperature mostly slightly decreased (Figure 5). The lowest contraction values were found for CRYO_spr_5 where a combination of amine catalysts Polycat 218 and Polycat 203 was used.

As a result of such changes in elongation at break and contraction, their ratio and, consequently, the safety coefficient (Figure 6) increased with the increase in the isocyanate index due to the more complete curing of PUR. The highest safety coefficient values were found for the foams CRYO_spr_4 and CRYO_spr_5, where the bismuth-based catalyst Dabco MB-20 was used together with the reactive amine catalyst Polycat 218 or a combination with Polycat 203.

FTIR spectroscopy (Figure 7) was used to confirm that the urethane formation reaction was complete and no free NCO groups remained. The peak of the stretching vibration at 2260–2280 cm^−1^ of free isocyanate was practically absent, in contrast to [54,55] where the isocyanate was predominant and not fully reacted. At the same time, all the vibrations characteristic of urethane groups are visible in the FTIR spectra in our fully reacted compositions CRYO_spr_2–CRYO_spr_5, such as 3300–3330 cm^−1^, which are the result of symmetric and asymmetric stretching vibrations of the N-H groups present in the urethane groups; the peaks at ~1520 cm^−1^ and ~1310 cm^−1^ are attributed to the in-plane N-H bending and NCO stretching of the urethane group [54] and others.

These two polyurethane foam compositions (CRYO_spr_4 and CRYO_spr_5) were found to be more suitable with a density of 35 kg/m^3^ for cryogenic application. The properties of the mentioned foams with reactive amine catalyst and bismuth-based Dabco MB-20 at cryogenic temperature were better than those of the foam with conventional additive amine catalyst Polycat 5 and tin-based catalyst Kosmos 19.

## 4. Conclusions

Low-density cryogenic foams with the environmentally friendly blowing agent hydrofluoroolefin (HFO) Solstice^®^ LBA were developed, prepared, and up-scaled. For the PUR foam preparation, new, low-toxicity, bismuth-based, recently developed and patented, low-emission reactive amine catalysts were used. Catalysts specially designed for blowing agents with low global warming potential were effective on such important cryogenic properties as adhesion strength after cryoshock and safety coefficient. Using the relatively HFO-stable blowing reactive amine catalyst in optimal concentrations, it is possible to obtain foam with a density lower than 36 kg/m^3^ and at the same time with a bond strength after cryoshock higher than 0.3 MPa and a safety coefficient higher than 3.9.

Our research shows that by combining more environmentally friendly catalysts and blowing agents it is possible to obtain PUR material, which in the future can serve as cryogenic insulation in liquefied natural gas (LNG) transportation or in space technologies.

## Figures and Tables

**Figure 1 materials-16-02798-f001:**
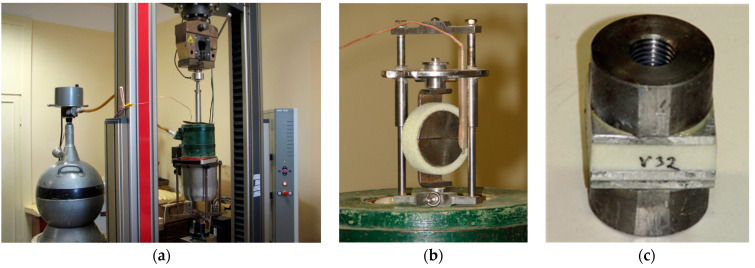
(**a**) Testing machine Zwick/Roell Z100 with a cryogenic test facility. (**b**) Appliance for ring tensile test at cryogenic temperature. (**c**) PUR sample for bond strength determination with adapters.

**Figure 2 materials-16-02798-f002:**
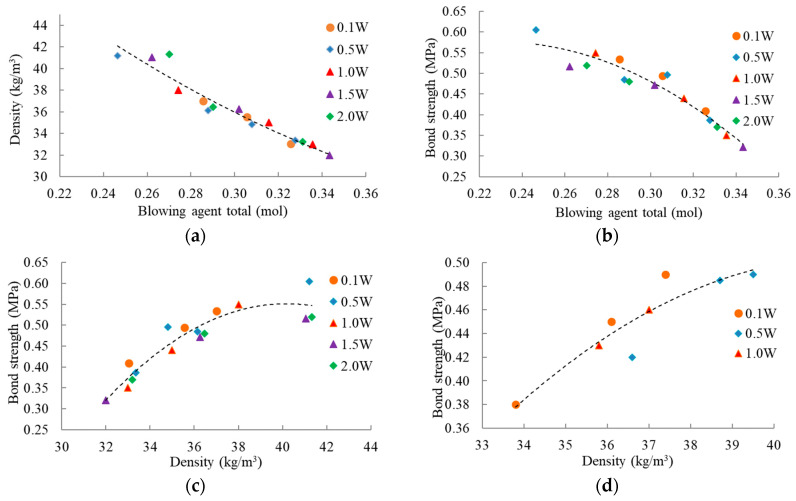
Relationships among total content of blowing agent, density, and bond strength. (**a**) Density of PUR foam vs. total content of blowing agents. (**b**) Tensile bond strength of PUR foam vs. total content of blowing agents. (**c**) Tensile bond strength of PUR foam vs. foam density. (**d**) Tensile bond strength of PUR foam after cryoshock test vs. foam density (water content in pbw in compositions in figure’s legend as: 0.1, …, 2.0 W).

**Figure 3 materials-16-02798-f003:**
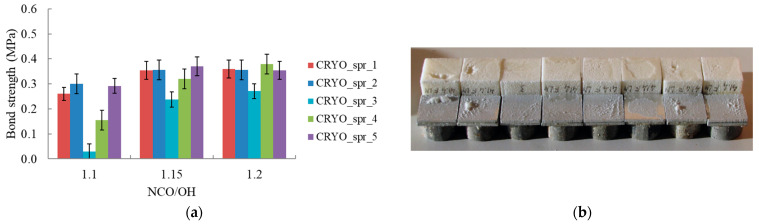
(**a**) Tensile bond strength of sprayed PUR foams after cryoshock test vs. NCO/OH ratio. (**b**) Sprayed PUR samples after cryoshock and tensile bond strength test.

**Figure 4 materials-16-02798-f004:**
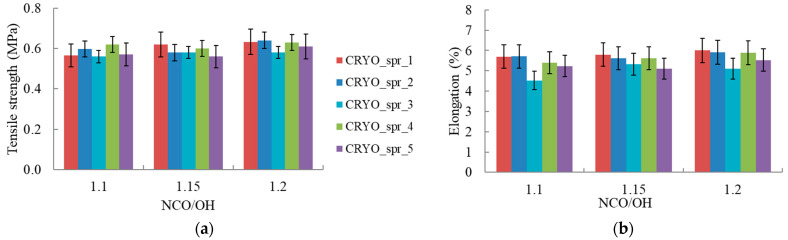
PUR foam tensile properties at 77 K vs. NCO/OH ratio: (**a**) tensile strength; (**b**) elongation at break.

**Figure 5 materials-16-02798-f005:**
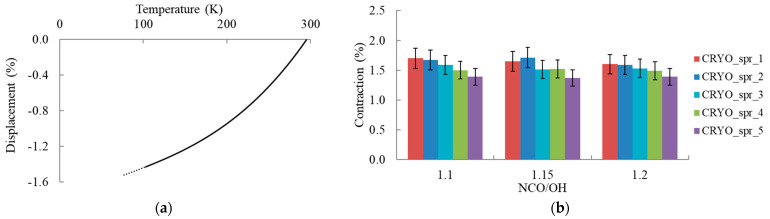
(**a**) Displacement curves measured by TMA test equipment with approximation; (**b**) PUR foam contraction at cooling from 275 to 77 K vs. NCO/OH ratio.

**Figure 6 materials-16-02798-f006:**
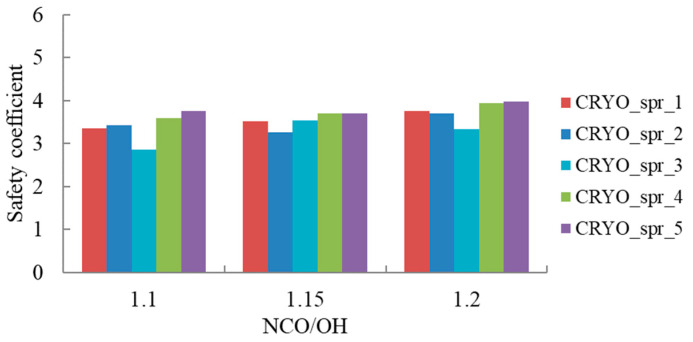
Safety coefficient of PUR foams at 77 K vs. NCO/OH ratio.

**Figure 7 materials-16-02798-f007:**
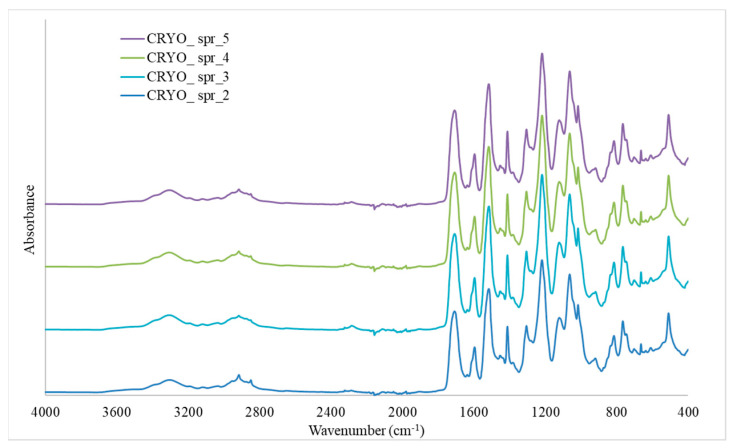
FTIR spectra of rigid PUR foam compositions.

**Table 1 materials-16-02798-t001:** Recipes of PUR pouring and spraying compositions, pbw.

	Ingredients	Trade Names	Pouring Composition	Spraying Composition
A-component	Polyols	Lupranol 3300	55	
		Lupranol 3508/1		
		Lupraphen 1901/1		
		Diethylene glycol	25	
		IXOL B 251	20	
	Flame retardant	TCPP	15	
	Surfactant	Silicone L-6915LV	1.5	
	Catalyst package		According to Table 2	
	Blowing agents	Solstice^®^ LBA	23–41	45
		Water	0–1.9	0
B-component (pMDI)		Desmodur^®^ 44V20L	147	147–160

**Table 2 materials-16-02798-t002:** Catalyst package and sample codes for PUR compositions.

Sample Codes	Catalyst, pbw				
	Kosmos 19	Dabco MB-20	Polycat 5	Polycat 218	Polycat 203
	Pouring compositions				
CRYO_p	0.1		0.5		
	Spraying compositions				
CRYO_spr_1	0.1		6		
CRYO_spr_2		0.15	6		
CRYO_spr_3		0.15		6	
CRYO_spr_4		0.2		6	
CRYO_spr_5		0.2		4	2

**Table 3 materials-16-02798-t003:** Content of blowing agents in moles used in pouring tests.

Water			Solstice^®^ LBA						
Added water	Water together with moisture of polyols		pbw						
			21	23	28	33	35	39	41
			moles						
pbw	pbw	moles	0.16	0.18	0.22	0.26	0.27	0.30	0.32
			Blowing agents total, moles						
0	0.1	0.01	*	*	0.23	*	0.28	0.31	0.33
0.4	0.5	0.03	*	*	0.25	0.29	0.30	0.33	0.35
0.9	1	0.06	*	0.24	0.28	0.32	0.33	*	*
1.4	1.5	0.08	*	0.26	0.30	0.34	0.35	*	*
1.9	2	0.11	0.27	0.29	0.33	*	*	*	*

* Compositions were not sprayed.

**Table 4 materials-16-02798-t004:** Foaming parameters.

Sample Codes	t_cream_	t_gel_	t_tack-free_
	Sec		
CRYO_p	23–27	38–45	45–60
CRYO_spr_1	3.4	11.0	17.0
CRYO_spr_2	3.5	11.5	19.8
CRYO_spr_3	3.6	14.0	23.2
CRYO_spr_4	3.7	13.2	22.2
CRYO_spr_5	4.5	15.0	27.1

**Table 5 materials-16-02798-t005:** Apparent densities (kg/m^3^) of spraying PUR compositions.

PUR Compositions	NCO/OH		
	1.1	1.15	1.2
CRYO_spr_1	34.6	34.8	35.3
CRYO_spr_2	34.3	34.7	34.9
CRYO_spr_3	34.4	34.9	35.1
CRYO_spr_4	34.5	34.6	35.0
CRYO_spr_5	34.5	34.6	35.3

## Data Availability

The data presented in this study are available on request from the corresponding author.

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
