# Peer review of "Influence of Reactive Amine-Based Catalysts on Cryogenic Properties of Rigid Polyurethane Foams for Space and On-Ground Applications"

_materials, 2023, doi:10.3390/ma16072798_

Round 1
Reviewer 1 Report
Authors of materials 2209505 reported results of experimental studies for the application of rigid polyurethane foams for cryogenic insulation. Revision in the following aspects are recommended before the publication in materials;
1) Most experimental results were attributed to the more complete cure. Evidence for the complete cure such as FT-IR data showing less isocyanate concentration or change of modulus values should be given.
2) Effects of amine catalysts were investigated in this study. It is expected that the reactivity such as gel times are affected by the catalyst type. Discussions on the reactivity resulting from the change of the amines are necessary.
3) Discussions on how amine catalysts affect the performance of the foams should be included to help readers to understand the scientific nature of the experimental results.
Reviewer 2 Report
This paper reports the effect of a reactive amine catalyst on the low temperature performance of polyurethane rigid foam. However, this paper lacks research depth in terms of mechanism. The author obtained samples with excellent performance by adjusting the catalyst formula. This mode is more suitable for applying for a patent rather than publishing an academic article, unless there are some in-depth research on mechanism. Therefore, I do not recommend the publication of the article.
Reviewer 3 Report
The authors describe the production, first at lab-scale and then upscaled, of different polyurethane foams, using different blends of blowing catalysts. The aim of this study is headed for their use in space aircrafts in order to have a good storage of hydrogen when used as aviation fuel, since the low temperatures at which it is kept during flight or travels can cause damages to the stocking units.
The paperi s well written, and the english is highly comprehensible, as well as the design of experiement and the characterization techniques.
However, maybe only few remarks may be assessed.
1. Despite the fact that the future perspectives of PU foams, in terms of sustainability, which is one of the main topics discussed in the introduction, stands in the use of non-isocyanate formulations, the number of studies in this field is a bit lower than those in which the use of bio-based polyols, obtained from renewable resources, is thoroughly discussed and also used for scaling-up processes. Some remarks about it in the introductive section might give a better picture of the environmental issue, for which the aerospace field is extremely sensitive. Here few examples:
· Sardon, H., Mecerreyes, D., Basterretxea, A., Averous, L., & Jehanno, C. (2021). From lab to market: current strategies for the production of biobased polyols. ACS Sustainable Chemistry & Engineering, 9(32), 10664-10677.
· Recupido, F., Lama, G. C., Ammendola, M., Bossa, F. D. L., Minigher, A., Campaner, P., ... & Verdolotti, L. (2023). Rigid composite bio-based polyurethane foams: From synthesis to LCA analysis. Polymer, 125674.
· Lin, C. S. K., Kirpluks, M., Priya, A., & Kaur, G. (2021). Conversion of food waste-derived lipid to bio-based polyurethane foam. Case Studies in Chemical and Environmental Engineering, 4, 100131.
2. In Figure 2, the values of the “Blowing agent total” in the graphs in a) and b), are not corresponding to those present in the Table 3.
3. It might be useful for the reader if the authors widen the range on both the abscissa and ordinate axes in Figure 2 for all the graphs.
4. The authors should give indications on the thermal conductivity of the studied materials.
Round 2
Reviewer 1 Report
Comments on the manuscript was reflected in the revision.
Author Response
Thank you for your review! and thank you, that you accept our corrections!
Reviewer 2 Report
In order to reflect the scientific nature of the article, the author has added some FTIR tests and conducted amine catalyst mechanism has been supplemented to some extent. However, there are still some issues that need to be supplemented and improved before the article is published.
1. In 2.1. Materials section, which catalyst is the "Reactive Amine Based Catalysts" mentioned in the title should be clearly explained.
2. In 2.1. Materials section, Can the approximate chemical structure of these "Reactive Amine Based Catalysts" be given?
3. The conclusion mentioned a lot about the influence of the fourth generation foaming agent on the foam performance, but the topic did not seem to mention the fourth generation foaming agent.
4. How to measure the density of foam? Is the density mentioned in the text an apparent density?
